# Peak expiratory flow rate and sarcopenia risk in older Indonesian people: A nationwide survey

Edi Sampurno Ridwan[1,2], Bayu Satria Wiratama[3,4], Mei-Yu Lin[5], Wen-Hsuan Hou[2,6,7,8,9], Megan Fang Liu[8], Ching-Min Chen[10], Hamam Hadi[11], Maw Pin Tan[12], Pei-Shan Tsai[2,13,14]*

1 Department of Nursing, Faculty of Health Sciences, Universitas Alma Ata, Yogyakarta, Indonesia, 2 School of Nursing, College of Nursing, Taipei Medical University, Taipei, Taiwan, 3 Graduate Institute of Injury Prevention and Control, College of Public Health, Taipei Medical University, Taipei, Taiwan, 4 Department of Biostatistics, Epidemiology and Population Health, Faculty of Medicine, Public Health and Nursing, Universitas Gadjah Mada, Yogyakarta, 5 Department of Nursing, Tzu Chi University of Science and Technology, Hualien, Taiwan, 6 Master Program in Long-Term Care, College of Nursing, Taipei Medical University, Taipei, Taiwan, 7 Department of Physical Medicine and Rehabilitation, Graduate Institute of Clinical Medicine, College of Medicine, Taipei Medical University, Taipei, Taiwan, 8 School of Gerontology Health Management, College of Nursing, Taipei Medical University, Taipei, Taiwan, 9 Center of Evidence-Based Medicine, Department of Education, Taipei Medical University Hospital, Taipei, Taiwan, 10 Department of Nursing, College of Medicine, National Cheng Kung University, Tainan, Taiwan, 11 Department of Nutrition, Faculty of Health Sciences, Universitas Alma Ata, Yogyakarta, Indonesia, 12 Department of Medicine, Faculty of Medicine, University of Malaya, Kuala Lumpur, Malaysia, 13 Department of Nursing and Center for Nursing and Healthcare Research in Clinical Practice Application, Wan Fang Hospital, Taipei Medical University, Taipei, Taiwan, 14 Sleep Science Center, Taipei Medical University Hospital, Taipei Medical University, Taipei, Taiwan

* ptsai@tmu.edu.tw

**Data Availability Statement:** All dataset files are available from the https://www.rand.org/well-being/social-and-behavioral-policy/data/FLS/IFLS.html.

## Abstract

Reduced peak expiratory flow is a common physiological change in older individuals and age is an important predictor for sarcopenia. We analyzed nationwide survey data to determine the relationship between peak expiratory flow rate and sarcopenia in older Indonesians. Community-dwelling Indonesian individuals aged ≥60 years (n = 2422; mean age = 67.21 y) from the fifth-wave data of the Indonesian Family Life Survey was selected. Sarcopenia was diagnosed based on handgrip strength, gait speed, and appendicular skeletal muscle mass measurements. Peak expiratory flow rates (PEFRs) were categorized according to their percentage of predicted flow rates as <50%, 50% to 80%, and >80%. Confounders previously determined to be associated with sarcopenia occurrence were included. Sarcopenia prevalence was 50.25%. After adjustment for confounders, PEFRs of <50% and 50% to 80% were associated with an increased sarcopenia risk (odds ratio = 5.22 and 1.88, respectively) compared with PEFRs of >80%. Poor lung function was independently associated with sarcopenia occurrence. Future studies should explore the usefulness of PEFR as a risk factor of sarcopenia.

**Funding:** This research did not receive any specific grant from funding agencies in the public, commercial, or not-for-profit sectors.

**Competing interests:** The authors have declared that no competing interests exist.

## Introduction

Sarcopenia is a degenerative muscle disease or muscle failure that occurs mainly in older individuals [1]. This primarily geriatric syndrome is characterized by loss of skeletal muscle mass and function, which can be defined based on diminished muscle strength and functional performance [1, 2]. The global prevalence of sarcopenia in individuals aged ≥60 years is 10% in both men and women, with a higher prevalence in non-Asian populations than Asian populations in both sexes [3]. Sarcopenia prevalence in individuals aged ≥50 years was 1% to 29% in community-dwelling populations, 14% to 33% in long-term care populations, and 10% in the acute hospital-care population, based on criteria defined by the European Working Group on Sarcopenia in Older People (EWGSOP) [4]. The Asia Working Group for Sarcopenia (AWGS) reported that estimated sarcopenia prevalence in Asia was 4.1% to 11.5% [2]. A prevalence meta-analysis revealed that the prevalence of sarcopenia in Japanese community-dwelling older people was 9.9%, based on the AWGS diagnostic criteria [5]. Sarcopenia prevalence in middle-aged and older individuals is substantial worldwide, although regional and age-associated variations in prevalence rates exist. Therefore, identifying factors associated with the occurrence of sarcopenia is crucial.

Sarcopenia is a major risk factor for frailty. Meta-analyses have revealed that sarcopenia is associated with increased rates of mortality, functional decline, falls, and hospitalizations [6]. Furthermore, individuals with sarcopenia have reduced quality of life compared with individuals without sarcopenia because of impaired physical function [7]. Regarding the economic burden of sarcopenia, the total cost of hospitalizations for patients with sarcopenia is higher than that for patients without sarcopenia [8]. This evidence confirms the risks and consequences of sarcopenia, thereby, indicating the need for improved prevention, screening, diagnosis, and management of this reportable medical condition [9].

Sarcopenia is an age-related disease. Aging is characterized by a decline in age-related metabolism and changes in cellular physicochemical properties, causing impaired self-regulation, impaired regeneration, structural changes, and impaired tissue and organ function, all of which are associated with the development of sarcopenia [10]. Therefore, reduced respiratory airflow is a common physiological change in older individuals [11]. Forced expiratory volume in 1 second ($FEV_1$), forced vital capacity (FVC), and peak expiratory flow rate (PEFR) decline with age [12], where age is an important predictor for sarcopenia [13]. Hence, the relationship between reduced respiratory airflow and sarcopenia warrants investigation. Furthermore, the population is aging rapidly in Indonesia, accounting 9,6% of the total population in 2019 [14]. The fast-growing aging population in Indonesia may cause a considerable public health burden in the near future because of an increase in reduced lung function and the sarcopenia cases. Therefore, because the association between reduced PEFR and sarcopenia should be explored in a nationally representative sample, this study examined whether reduced PEFR independently predicted sarcopenia in community-dwelling older Indonesians after adjustments for confounders.

## Materials and methods

### Study sample and procedure

Survey data from the fifth wave of the Indonesian Family Life Survey (IFLS-5) was analyzed. The IFLS is a large-scale longitudinal survey that collects data on the same individuals at multiple time points, as well as an extensive array of retrospective information, to provide insights into the dynamic relationship between the individual, household, and community [15]. The survey examined individuals living in households from 13 of the 27 provinces of Indonesia,

representing 83% of the population. The population was stratified in provinces and urban or rural areas. Sample survey selected using random sampling within the strata to maximize the representativeness of the population, capture the diversity of culture and socioeconomic status, and maximize the cost-effectiveness of the survey. In the IFLS-5, 92% and 90.5% of dynasty households (ie, households included in IFLS-1) and individual target households were recontacted, respectively. Data from 16 204 households and 58 312 individuals were extracted from the IFLS-5. The IFLS-5 was approved by the Institutional Review Board of the Survey METER in Indonesia and Research and Development, Indonesia (RAND). The protocol approval number was S0064-06-01-CR01 [15].

Individuals aged ≥60 years who had completed the IFLS-5 interview were selected to be included in this study. Those who had missing data were excluded. In total, 5363 respondents aged 60 years were identified and 3962 completed interviews. The final analysis was performed on data from 2422 respondents after the exclusion of those with missing data (n = 1540; Fig 1). The selected sample was weighted for national representativeness in the cross-sectional analysis.

## Sarcopenia

Sarcopenia was defined as a progressive reduction in age-related skeletal muscle mass and function as determined by muscle strength and physical performance. Sarcopenia was diagnosed based on the AWGS criteria; Fig 2 [2]. Participants with normal handgrip strength, gait speed, and muscle mass were classified as individuals without sarcopenia, whereas individuals with low handgrip strength, gait speed, and muscle mass were classified as individuals with sarcopenia. A prediction equation based on age, sex, height, and weight was used to determine appendicular skeletal muscle mass (ASM) [16]. ASMs of <23.4 and <15.5 kg were classified as low muscle mass in men and women, respectively [17]. The handgrip strength of the dominant hand was measured 2 times (in kg) by using a Baseline Smedley spring-type dynamometer (Baseline Dianometer, NY, USA) in a standing position. Handgrip strength was classified as low if it was <26 kg for men and <18 kg for women [2]. Gait speed was determined based on a timed walk on a 2.5-m straight flat surface. If gait speed was <0.8 m/s, it was categorized as low [2, 18].

## PEFR

PEFR was defined as the maximum flow rate that could be achieved by a forced expiratory maneuver [19]. PEFR was measured using a Vitalograph peak flow meter, and the results were recorded if respondents demonstrated maximal effort during the test. Predicted PEFRs were calculated and the measured PEFR were categorized based on their percentage of the predicted PEFR into <50%, 50% to 80%, and > 80%. A PEFR of <80% of the predicted value indicates possible airflow obstruction [20].

## Confounders

Confounders previously determined to be associated with sarcopenia occurrence were included [1, 2, 6, 21–23]. These factors were age, sex, urbanization level, smoking status, body mass index (BMI), lower limb strength (LLS), physical activity, fall history, sleep quality, depressive symptoms, history of hypertension, diabetes mellitus, COPD, cardiovascular disease, and rheumatoid arthritis. The urbanization level was categorized as rural or urban and was determined by population density, percentage of agricultural households, and distance to urban facilities. We assessed smoking status based on the questions: "Have you ever chewed tobacco, smoked a pipe, smoked self-rolled cigarettes, or smoked cigarettes or cigars?" "Do

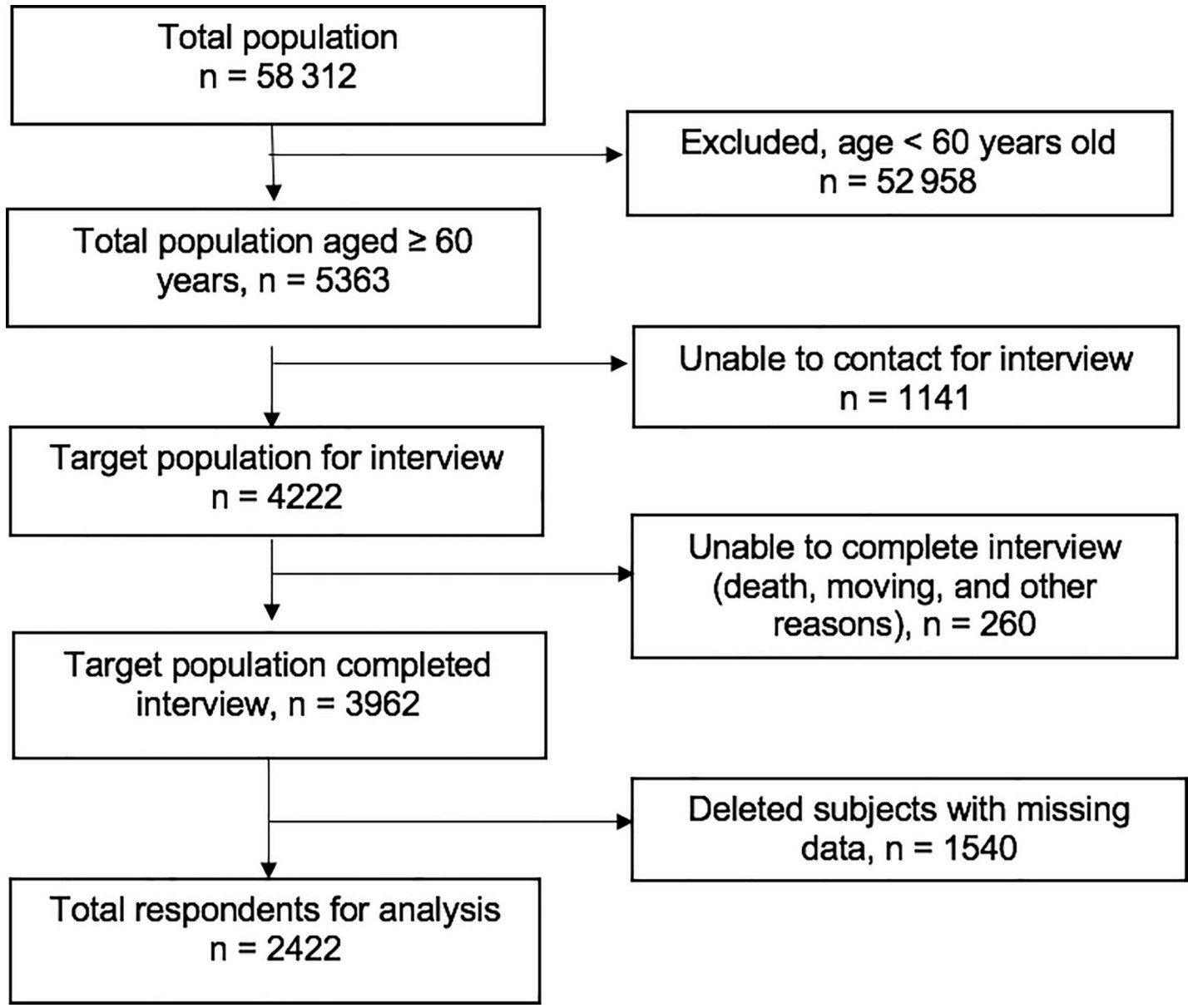

**Fig 1. Flow diagram for sample selection process included respondents aged 60 and over that completed interview and had complete data.**

you still have the habit or have you quit completely (still have the habit or quit)?" The responses were categorized as 0 = nonsmoker, 1 = current smoker, and 2 = past smoker. BMI was recorded as both a continuous and a categorical variable. BMI was categorized into underweight ($\leq$18.5 kg/m$^2$), normal weight (18.6–22.9 kg/m$^2$), and overweight ($\geq$23.0 kg/m$^2$) based on the World Health Organization's recommended values for Asia [24]. LLS was calculated as the mean of 5 repeated chair stand tests. The data were grouped into low strength (= 0) and normal strength (= 1) based on published guidelines [25]. The level of physical activity was determined for each participant based the responses to the International Physical Activity Questionnaire (IPAQ), which includes questions regarding the time spent on different types of physical activity over the last 7 days [26]: "Report about the vigorous-intensity activities you

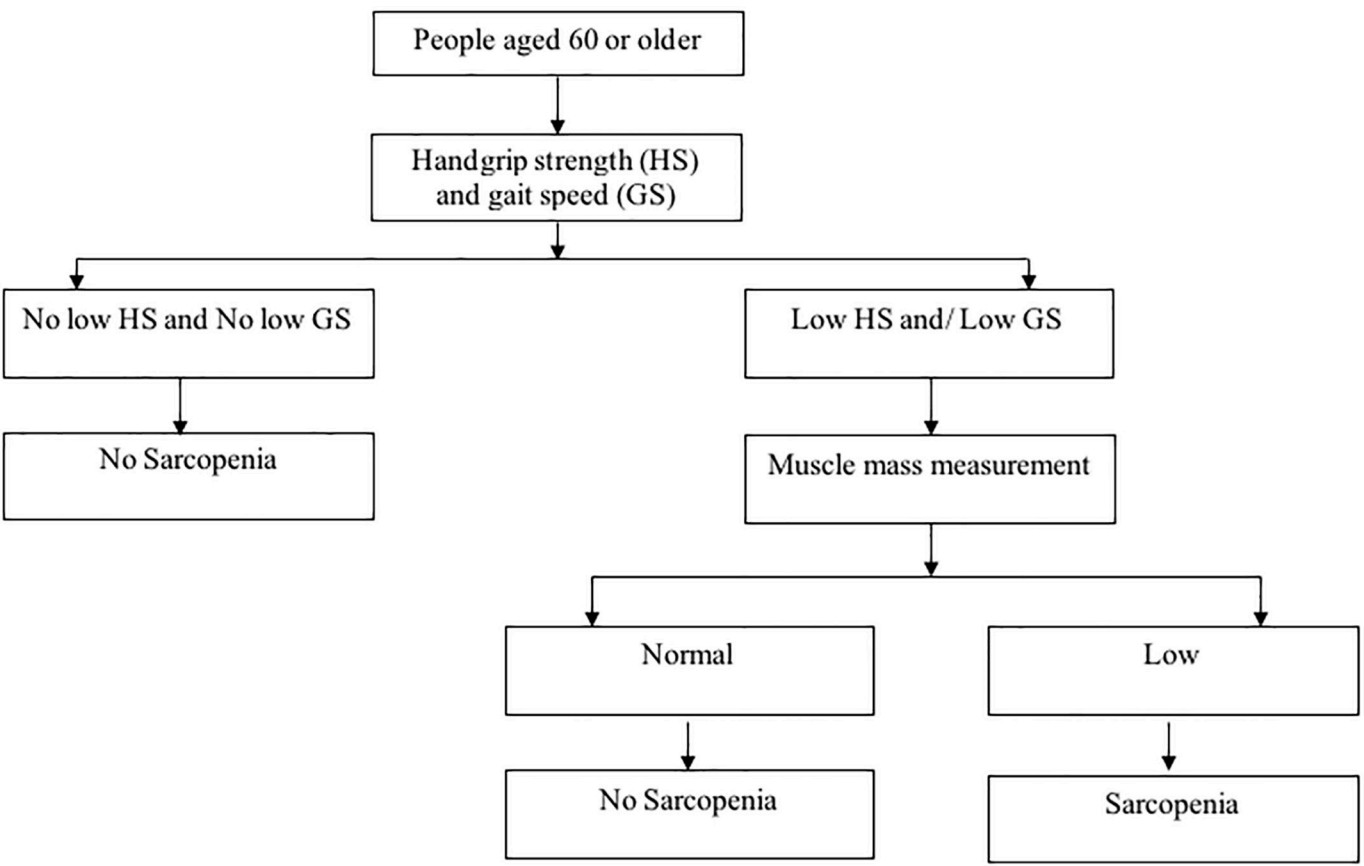

**Fig 2. Algorithm for diagnosing sarcopenia adopted from Asia Working Group for Sarcopenia** [2].

have done in the past 7 days." "Report about the moderate-intensity activities you have done in the past 7 days." "Report about the time that you have spent walking in the past 7 days." The responses were categorized as high = 2 if achieving minimum of at least 3000 MET-minutes per week, moderate = 1 if achieving minimum of at least 600 MET-minutes per week, and low = 0 level of physical activity if no activity reported. Fall history was assessed based on the following questions: "Have you fallen down in the last 2 years (yes or no)?" "How many times have you fallen down in the last 2 years?" The responses were categorized as no falls = 0, single fall (1 or 2 falls) = 1, and multiple falls (3 or more) = 2. Sleep quality was determined based on the responses to 5 questions on sleep quality from the Patient-Reported Outcomes Measurement Information System (PROMIS-10) which had a Cronbach's alpha for internal consistency of 0.9 [27]. Participants were asked to recall their sleep quality over the past 7 days, and each question was scored into 3 response categories: 0 = poor, 1 = fair, and 2 = good quality of sleep. Depressive symptoms were assessed based on 10 questions from the short version of the Center for Epidemiologic Studies-Depression (CES-D) scale which had a Cronbach's alpha of 0.87 for internal consistency [28]; patients were classified as 0 = nondepressive and 1 = depressive, with a cutoff score of 10. Incident chronic diseases were determined based on the answer ("yes" or "no") to the following question "Has a doctor, paramedic, nurse, or midwife ever diagnosed you as having [. . .]?" The answers were coded as 0 = no disease and 1 = disease occurrence.

## Statistical analysis

Continuous variables with normal distribution are presented as means and standard deviation, and categorical variables as frequencies and percentages. The significance level (alpha) for hypothesis testing was set at $P < .05$. The bivariate association between each confounder and sarcopenia was assessed using univariate binomial logistic regression analysis. Multivariate binomial logistic regression analysis was performed to examine the predictive ability of PEFR for sarcopenia risk after adjustment for confounders. Variables with $P < .25$ in the bivariate analysis were included in the multivariate analysis [29]. A forward and stepwise logistic regression method was applied to identify the effects of confounders in each model. The results of the pairwise correlations of variances were used to identify multicollinearity among confounders. Post estimation analyses were performed to determine whether the models fit the measured outcome, and Pearson or Hosmer-Lemeshow goodness-of-fit tests were performed. Sampling weights were constructed for cross-sectional analysis to ensure the national representativeness of the estimate. Data were analyzed using STATA (version 14) [30].

## Results and discussion

In total, 2422 individuals were included in this study, of which 1213 (50.08%) were women and 1209 (49.92%) were men, with a mean age of 67.21 (SD = 6.15) years. Of all included individuals, 1217 (50.25%) individuals met the study criteria for the diagnosis of sarcopenia, with 654 (53.92%) of 1213 women and 563 (46.57%) of 1209 men having received a diagnosis of sarcopenia. Individuals with a PEFR <50%, 50% to 80%, and >80% of the predicted value were 1690 (69.78%), 540 (22.30%), and 192 (7.93%), respectively.

Table 1 presents the distribution of demographic characteristics, the confounders, and the association of each confounder with sarcopenia, analyzed using simple logistic regression analyses. The sociodemographic variables that were significantly associated with sarcopenia were age ($P < .001$), sex ($P < .001$), and urbanization level ($P < .001$), whereas smoking status was not correlated to sarcopenia ($P = .54$). Furthermore, the physical factors correlated to sarcopenia were BMI, LLS, and physical activity ($P < .001$, $P < .001$, and $P = .007$, respectively), whereas the history of falls was not ($P = .08$). Regarding psychological factors, sleep quality was significantly associated with sarcopenia, whereas depressive symptoms were not ($P = .052$). Moreover, none of the chronic diseases, such as hypertension ($P = .92$), diabetes mellitus ($P = .35$), COPD ($P = .95$), cardiovascular disease ($P = .32$), and rheumatoid arthritis ($P < .058$), was significantly associated with sarcopenia.

The results of the multivariate binomial logistic regression analysis with sarcopenia as the outcome measure are presented in Table 2. A PEFR of <50% and 50% to 80% independently predicted the likelihood of sarcopenia after adjustment for confounders (odds ratios [ORs] = 5.22 and 1.88, respectively). The goodness-of-fit test also indicated a good model fit ($P = .28$).

## Discussion

Poor lung function indicated by PEFR of <80% was independently associated with sarcopenia occurrence in older Indonesians: PEFRs of <50% and 50% to 80% were associated with nearly 5.22 and 1.88 times increased sarcopenia risk, respectively. In this study, sarcopenia was predicted by reduced respiratory airflow determined by PEFR.

Studies have reported that the cause of sarcopenia is multifactorial [1, 2]. Our analysis indicated that lower PEFR or the presence of obstructive airway increased the probability of sarcopenia in older community-dwelling Indonesians. According to Jeon et al., low pulmonary function was significantly associated with low muscle mass in older community-dwelling people with COPD where low muscle mass is an indicator of sarcopenia [31]. Jeon *et al* revealed

**Table 1. Bivariate analysis of demographic characteristics and confounders associated with sarcopenia.**

| Variables | All Participants (n = 2422) | Nonsarcopenic (N = 1205) | Sarcopenic (N = 1217) | P |
|---|---|---|---|---|
| Age, *mean (SD)* | 67.21 (6.15) | 65.53 (5.07) | 68.86 (6.67) | <.001 |
| Sex, *n (%)* | | | | |
| 0 = Female | 1213 (50.08) | 559 (46.39) | 654 (53.74) | <.001 |
| 1 = Male | 1209 (49.92) | 646 (53.61) | 563 (46.26) | |
| Living area, *n (%)* | | | | |
| 0 = Rural | 1138 (46.99) | 515 (42.74) | 623 (51.19) | <.001 |
| 1 = Urban | 1284 (53.01) | 594 (57.26) | 594 (48.81) | |
| Smoking habits, *n (%)* | | | | |
| 0 = Nonsmoker | 1321 (54.54) | 643 (53.36) | 678 (55.71) | 0.54 |
| 1 = Current smoker | 804 (33.20) | 418 (34.69) | 386 (31.72) | |
| 2 = Past smoker | 297 (12.26) | 144 (11.95) | 153 (12.57) | |
| BMI, *mean (SD)* | 22.38 (4.27) | 23.35 (4.40) | 21.41 (3.91) | <.001 |
| BMI category, *n (%)* | | | | |
| 1 = Underweight | 437 (18.04) | 144 (11.95) | 293 (24.08) | <.001 |
| 2 = Normal | 1004 (41.45) | 477 (39.59) | 527 (43.30) | |
| 3 = Overweight | 981 (40.50) | 584 (48.46) | 397 (32.62) | |
| LLS, *n (%)* | | | | |
| 0 = Low | 1018 (42.03) | 393 (32.61) | 625 (51.36) | <.001 |
| 1 = Normal | 1404 (57.97) | 812 (67.39) | 592 (48.64) | |
| Sleep quality, *n (%)* | | | | |
| 0 = Poor | 233 (9.62) | 116 (9.63) | 117 (9.61) | 0.036 |
| 1 = Fair | 1036 (42.77) | 549 (45.56) | 487 (40.02) | |
| 2 = Good | 1153 (47.61) | 540 (44.81) | 613 (50.37) | |
| Depressive symptoms, *n (%)* | | | | |
| 0 = No | 2054 (84.81) | 1039 (86.22) | 1015 (83.40) | 0.052 |
| 1 = Yes | 368 (15.19) | 166 (13.78) | 202 (16.60) | |
| Fall history, *n (%)* | | | | |
| 0 = No falls | 2297 (94.84) | 1153 (95.68) | 1144 (94.00) | 0.08 |
| 1 = Single fall | 75 (3.10) | 31 (2.57) | 44 (3.62) | |
| 2 = Multiple falls | 50 (2.06) | 21 (1.74) | 29 (2.38) | |
| Physical activity, *n (%)* | | | | |
| 0 = Low | 1673 (69.08) | 804 (66.72) | 869 (71.41) | 0.007 |
| 1 = Moderate | 255 (10.52) | 130 (10.79) | 125 (10.27) | |
| 2 = High | 494 (20.40) | 271 (22.49) | 223 (18.32) | |
| Predicted PEFR, *n (%)* | | | | |
| 0 = < 50% | 1,690 (69.75) | 709 (58.79) | 981 (80.61) | <.001 |
| 1 = 50–80% | 540 (22.33) | 344 (28.52) | 197 (16.19) | |
| 2 = > 80% | 191 (7.92) | 152 (12.69) | 39 (3.20) | |
| Hypertension, *n (%)* | | | | |
| 0 = No | 1755 (72.46) | 872 (72.37) | 883 (72.56) | 0.92 |
| 1 = Yes | 667 (27.54) | 333 (27.63) | 334 (27.44) | |
| Diabetes mellitus, *n (%)* | | | | |
| 0 = No | 2280 (94.14) | 1129 (93.69) | 1151 (94.58) | 0.35 |
| 1 = Yes | 142 (5.86) | 76 (6.31) | 66 (5.42) | |
| COPD, *n (%)* | | | | |
| 0 = No | 2306 (95.21) | 1147 (95.19) | 1159 (95.23) | 0.95 |

*(Continued)*

**Table 1.** (Continued)

| Variables | All Participants (n = 2422) | Nonsarcopenic (N = 1205) | Sarcopenic (N = 1217) | P |
|---|---|---|---|---|
| 1 = Yes | 116 (4.79) | 58 (4.81) | 58 (4.77) | |
| Cardiovascular, *n (%)* | | | | |
| 0 = No | 2323 (95.91) | 1151 (95.52) | 1172 (96.30) | 0.32 |
| 1 = Yes | 99 (4.09) | 54 (4.48) | 45 (3.70) | |
| Rheumatoid arthritis, *n (%)* | | | | |
| 0 = No | 2105 (86.91) | 1063 (88.22) | 1042 (85.62) | 0.058 |
| 1 = Yes | 317 (13.09) | 142 (11.78) | 175 (14.38) | |

BMI, body mass index; COPD, chronic obstructive pulmonary disease; PEFR, peak expiratory flow rate

that subjects with low $PEV_1$ and FVC had a lower average muscle mass compared with subjects with normal $PEV_1$ and FVC. Similarly, previous studies indicated that $FEV_1 < 80\%$, $FVC < 80\%$, and $PEF < 80\%$ were associated with reduced skeletal muscle mass determined by handgrip strength and chair stand testing [32, 33]. Reduced pulmonary function could be a result of low muscle strength or muscle mass. Forced expiratory volume in 1 second ($FEV_1$) was significantly lower in subjects with sarcopenia and their $FEV_1$ rate declined faster than in subjects without sarcopenia [34]. A study showed that PEFR in patients with sarcopenia were lower compared to non-sarcopenics [35]. Moreover, Shin *et al* reported that individuals with low handgrip strength exhibited lower maximal inspiratory pressure and maximal expiratory pressure because reduced respiratory muscle strength is associated with a reduced upper extremity muscle strength [36]. It has been demonstrated that PEFR significantly correlated to

**Table 2. Multivariate binomial logistic regression model of predictors of sarcopenia.**

| Confounders | OR | SE | Z | P | 95% CI |
|---|---|---|---|---|---|
| Age | 1.062 | 0.009 | 7.26 | <. 001 | 1.045–1.079 |
| Sex | | | | | |
| 1 = Male | 1.48 | 0.184 | 3.16 | 0.002 | 1.161–1.888 |
| 0 = Female (Ref.) | 1 | 1 | 1 | 1 | 1 |
| BMI, | | | | | |
| 1 = Underweight | 1.761 | 0.226 | 4.41 | <. 001 | 1.369–2.265 |
| 2 = Normal (Ref) | 1 | 1 | 1 | 1 | 1 |
| 3 = Overweight | 0.66 | 0.066 | −4.17 | <. 001 | 0.542–0.802 |
| LLS | | | | | |
| 0 = Low | 1.73 | 0.164 | 5.78 | <. 001 | 1.436–2.083 |
| 1 = Normal (Ref) | 1 | 1 | 1 | 1 | 1 |
| Pred. PEFR | | | | | |
| <50% | 5.224 | 1.115 | 7.74 | <. 001 | 3.437–7.938 |
| 50%-80% | 1.887 | 0.39 | 3.07 | 0.002 | 1.258–2.830 |
| 80% (Ref) | 1 | 1 | 1 | 1 | 1 |
| Physical activity | | | | | |
| 0 = Low | 1.263 | 0.142 | 2.08 | 0.037 | 1.013–1.573 |
| 1 = Moderate | 1.34 | 0.227 | 1.72 | 0.085 | 0.960–1.868 |
| 2 = High | 1 | 1 | 1 | 1 | 1 |

BMI, body mass index; LLS, lower limb strength; PEFR, peak expiratory flow rate; OR, odds ratio; SE, standard error; 95% CI, confidence interval.

handgrip strength, gait speed, and maximal inspiratory pressure [37]. PEFR may also be an indicator of respiratory muscle strength and has been used to identify sarcopenia [38, 39]. Ohara *et al* and Kera *et al* confirmed that PEFR could be used to determine the presence of sarcopenia or as a valid indicator for sarcopenia, with areas under receiver operating characteristic curves of 0.70 and 0.73, respectively [38, 39]. These results concurred with our finding that PEFR remained an independent predictor of the occurrence of sarcopenia after adjustments for LLS and physical activity.

We used reduced PEFR to predict sarcopenia occurrence. PEFR is a simple measure of the maximum speed of expiration to monitor a person's ability to exhale [20]. To determine the clinical significance of PEFR, peak flow scores were categorized into < 50%, which indicated serious airway obstruction, and <80%, which indicated airway obstruction [20]. Pothirat *et al* reported that PEFR could replace $FEV_1$ as a measure of COPD severity, although the agreement between these measures was not satisfactory [19]. These findings indicate the usefulness of the measure in monitoring progressive airway limitations. Furthermore, the patent airway is essential for tissue oxygenation because tissue hypoxia, which results from inadequate oxygen transport, causes impairments of muscle function and reduces myotube size in tissue-engineered skeletal muscle [40]. Multifactorial cellular mechanisms exacerbate muscle atrophy and loss of muscle strength (which occurs primarily in patients with COPD). PEFR measures can be personalized, simple, and affordable. Therefore, measuring PEFR could be the first step in detecting sarcopenia in low-income countries, such as Indonesia

On the basis of current cutoff criteria for sarcopenia, half of Indonesians aged ≥60 years were sarcopenic. Han *et al* reported that sarcopenia prevalence, based on the AWGS diagnosis criteria, in individuals aged ≥60 years in the Chinese suburb-dwelling population was 6.4% in men and 11.5% in women [41]. A meta-analysis indicated that sarcopenia prevalence among Asian individuals aged ≥60 years was 19% in women and 10% in men, with an overall global prevalence of 10% in both men and women [3]. Compared with these findings, sarcopenia prevalence in Indonesia is a public health concern. However, Kim *et al* reported that sarcopenia prevalence in a community-dwelling Japanese elderly population differed based on the cutoff values; sarcopenia prevalence of 2.5% to 28.0% in men and 2.3% to 11.7% in women when measured through dual-energy X-ray absorptiometry and 7.1% to 98.0% in men and 19.8% to 88.0% in women when measured using bioelectrical impedance analysis [42]. Adopting different consensus definitions affects the estimated sarcopenia prevalence; however, estimating the overall sarcopenia prevalence among the older population in Indonesia provided essential data, particularly when considering the growing aging population in the country. We identified sarcopenia as a highly prevalent condition among Indonesians aged ≥60 years, which supports the notion that sarcopenia is an age-related disease occurring substantially in older individuals [2, 4].

Our analysis revealed that both underweight and overweight, low limb strength, and low levels of physical activity were significantly and independently associated with sarcopenia. Studies have provided compelling evidence that obesity is a significant predictor for sarcopenia and that its effects are exerted through multiple pathways and mechanisms as a result of biological or biochemical dysfunction because of various lifestyle aspects that affect both obesity and sarcopenia pathophysiological processes [43]. Therefore, obesity has been identified as an attribute of the aging process through the complex interactions of various effecting factors, including neuromuscular transmission, muscle architecture, fiber composition, excitation–contraction coupling, and metabolisms involved in mechanisms related to muscle mass and muscle strength [44]. This complex process may explain the discrepancy in previous findings, wherein individuals with and without obesity exhibited sarcopenia [45]. Furthermore, multiple pathways underlie crucial changes in body composition, particularly BMI, fat percentage, and

LLS reduction [46]. Regarding psychosocial factors, our findings indicated that sleep quality and depressive symptoms were not predicting factors of sarcopenia occurrence, despite studies reporting a correlation between depressive symptoms and sarcopenia [21]. Although a meta-analysis reporting that sarcopenia is highly prevalent in individuals with CVD, diabetes mellitus, and respiratory diseases [47], our results indicated that chronic diseases such as diabetes mellitus, COPD, heart diseases, and rheumatoid arthritis were not independent predictors of sarcopenia occurrence in older Indonesians.

### Strengths and limitations

The strength of our study is that we used data from a nationally representative sample and established guidelines were adopted to identify sarcopenia.

This study has several limitations. First, muscle mass was determined by a surrogate measure which uses a prediction equation to determine the ASM. The available consensus criteria for sarcopenia according to the AWGS and EWGSOP recommend using dual X-ray absorptiometry and bioimpedance, but not an estimation equation. However, the equation has been validated with dual X-ray absorptiometry and bioimpedance [16]. Second, PEFR is more commonly employed to determine airway obstruction severity in individuals with asthma, and its accuracy may be patient dependent [20]. Third, because this was a cross-sectional study, a cause–effect relationship could not be inferred. Nevertheless, using data obtained from a middle-income developing nation, we revealed a role for PEFR as a biomarker for sarcopenia, which deserves further evaluation in prospective studies.

### Conclusions

Reduced PEFR is associated with the presence of sarcopenia in a representative sample of Indonesian residents aged ≥60 years. Future studies should further evaluate the potential of PEFR as a biomarker for sarcopenia as a low-cost and objective measurement tool in low-to-middle–income countries.

### Author Contributions

**Conceptualization:** Edi Sampurno Ridwan, Wen-Hsuan Hou, Megan Fang Liu, Ching-Min Chen, Hamam Hadi, Maw Pin Tan, Pei-Shan Tsai.

**Data curation:** Edi Sampurno Ridwan, Mei-Yu Lin, Pei-Shan Tsai.

**Formal analysis:** Edi Sampurno Ridwan, Bayu Satria Wiratama, Mei-Yu Lin, Pei-Shan Tsai.

**Investigation:** Edi Sampurno Ridwan, Wen-Hsuan Hou, Megan Fang Liu, Ching-Min Chen, Hamam Hadi, Maw Pin Tan, Pei-Shan Tsai.

**Methodology:** Edi Sampurno Ridwan, Pei-Shan Tsai.

**Project administration:** Edi Sampurno Ridwan, Pei-Shan Tsai.

**Software:** Edi Sampurno Ridwan, Bayu Satria Wiratama, Mei-Yu Lin, Pei-Shan Tsai.

**Supervision:** Edi Sampurno Ridwan, Wen-Hsuan Hou, Megan Fang Liu, Ching-Min Chen, Hamam Hadi, Maw Pin Tan.

**Visualization:** Edi Sampurno Ridwan, Pei-Shan Tsai.

**Writing – original draft:** Edi Sampurno Ridwan, Pei-Shan Tsai.

**Writing – review & editing:** Edi Sampurno Ridwan, Wen-Hsuan Hou, Megan Fang Liu, Ching-Min Chen, Hamam Hadi, Maw Pin Tan, Pei-Shan Tsai.

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
