## [Decision Letter · Decision Letter 0]

10 Nov 2020

PONE-D-20-24364

Peak expiratory flow rate and sarcopenia risk in older Indonesian people: A nationwide survey

PLOS ONE

Dear Dr. Ridwan,

Thank you for submitting your manuscript to PLOS ONE. After careful consideration, we feel that it has merit but does not fully meet PLOS ONE’s publication criteria as it currently stands. Therefore, we invite you to submit a revised version of the manuscript that addresses the points raised during the review process. Please pay close attention to each point raised by the two expert Reviewers as you revise your manuscript.

We look forward to receiving your revised manuscript.

Kind regards,

Stephen E Alway, Ph.D.

Academic Editor

PLOS ONE

Journal Requirements:

2. Please amend the manuscript submission data (via Edit Submission) to include authors Bayu Satria Wiratama, Mei-Yu Lin, Wen-Hsuan Hou, Megan Fang Liu, Ching-Min Chen, Hamam Hadi, Maw Pin Tan and Pai-Shan Tsai.

Reviewers' comments:

Reviewer's Responses to Questions

**Comments to the Author**

1. Is the manuscript technically sound, and do the data support the conclusions?

Reviewer #1: Yes

Reviewer #2: Yes

2. Has the statistical analysis been performed appropriately and rigorously? 

Reviewer #1: Yes

Reviewer #2: Yes

3. Have the authors made all data underlying the findings in their manuscript fully available?

Reviewer #1: Yes

Reviewer #2: No

4. Is the manuscript presented in an intelligible fashion and written in standard English?

Reviewer #1: Yes

Reviewer #2: Yes

5. Review Comments to the Author

Reviewer #1: Title: Title seems to be suitable for the manuscript.

Abstract: The abstract of the study is brief and understandable.

Introduction:

-The purpose of the article and the aim was expressed in the right way. Albeit, please shorten some parts of the introduction to make it more straightforward. Also, reduce the references that you cited.

-You may wish to cite current guideline of European Working Group on Sarcopenia in Older People (EWGSOP) instead of reference 1. (Sarcopenia: revised European consensus on definition and diagnosis)

Material and methods:

-Add the date and number of ethical approval.

-Authors mentioned that Gait speed was determined based on a timed walk on a 2.5-m straight flat surface. Please cite reference for this measurement. Do authors mean 4 or 6 meter walking speed?

- What are the inclusion and exclusion criteria of the study?

Results:

-What are the differences between sarcopenic and non-sarcopenic patients according to the PEFR <50%, 50% to 80%, and >80% of the predicted value? You had better write in table 1.

Discussion:

-You may wish to cite some more recent papers which support this manuscript’s findings (please see below-references section)

- Line 237 and 238 ----- ‘We also demonstrated that handgrip strength was significantly associated with FEV1, maximal voluntary ventilation, and lung function (37, 38).’ Did you demonstrate that in your study?

References:

-There is no reference from 2020. For an article that will be published this year, the authors may well wish to add some more recent papers (i.e. the very nice article on diaphragmatic muscle thickness and PEFR measurements in sarcopenic patients-Diaphragmatic muscle thickness in older people with and without sarcopenia, PMID: 32406014)

-Try to reduce the references

- Revise references, many of them are incorrect, including the abbreviations of the journals or non-sense words. See journal information for authors

Reviewer #2: This is a observational study pèrforme din a large sample of community-dwelling Indonesian individuals aged ≥60 years (n = 2422) aimed to analyses prevalenc eof sarcopenia and its relationship with peak flow rate as an useful biomarker for analysis of sarcopenia. There are some major cocenrns that need to be adressed in order to support the methodology and thus, the authors' conclusions.

Authors need to include and cite all study performed in ASIA about sarcopenia prevalence in older indivudals omly one reference has been cited (page 3 line 61).

Authors shoudl clarify how sarcopenia criteria were use d to diagnose sarcopenia because one individuals fulfilled one or two criteria out 3 what happens? an algorithm of diagnosis of sarcopenia should be included.

Sensitivity, specificity and alpha cronbach data should be stated for all questionnaire used in the study.

The criteria used to classified individuals as having "high", "moderate", and "low" physical activity are not explained in details, cut off values (minutes/per week) should be included and referenced properly a sit is a crucial variable in this study and in general in the sarcopenia research field.

Relevant study published on respiratory function and sarcopenia have not been cited, hence the discussion is quite poor:

Landi F, Salini S, Zazzara MB, Martone AM, Fabrizi S, Bianchi M, Tosato M, Picca A, Calvani R, Marzetti E. Relationship between pulmonary function and physical performance among community-living people: results from Look-up 7+ study. J Cachexia Sarcopenia Muscle. 2020 Feb;11(1):38-45.

Martínez-Arnau FM, Buigues C, Fonfría-Vivas R, Cauli O. Respiratory Muscle Strengths and Their Association with Lean Mass and Handgrip Strengths in Older Institutionalized Individuals. J Clin Med. 2020 Aug 24;9(9):2727.

Pacifico J, Geerlings MAJ, Reijnierse EM, Phassouliotis C, Lim WK, Maier AB. Prevalence of sarcopenia as a comorbid disease: A systematic review and meta-analysis. Exp Gerontol. 2020 Mar;131:110801

6. PLOS authors have the option to publish the peer review history of their article (what does this mean?). If published, this will include your full peer review and any attached files.

Reviewer #1: No

Reviewer #2: No

---

## [Author Response · Author response to Decision Letter 0]

23 Dec 2020

Responses to the reviewer’s comments:

1. Is the manuscript technically sound, and do the data support the conclusions? Reviewer #1: Yes; Reviewer #2: Yes.

Response:

I would like to thank the reviewer for the acknowledgement of this manuscript.

2. Has the statistical analysis been performed appropriately and rigorously? Reviewer #1: Yes; Reviewer #2: Yes.

Response:

Thank you very much for the positive comments on statistical analysis performance. 

3. Have the authors made all data underlying the findings in their manuscript fully available? Reviewer #1: Yes; Reviewer #2: No.

Response:

I would like to thank for the reviewer’s comments. I have presented the underlying data that was analyzed fully available.

4. Is the manuscript presented in an intelligible fashion and written in standard English? Reviewer #1: Yes; Reviewer #2: Yes.

Response:

Thank you very much for this constructive comments

Reviewer #1:

1. Title: Title seems to be suitable for the manuscript. Abstract: The abstract of the study is brief and understandable.

Response:

I appreciate this positive comment. 

2. Introduction:

The purpose of the article and the aim was expressed in the right way. Albeit, please shorten some parts of the introduction to make it more straightforward. Also, reduce the references that you cited. You may wish to cite current guideline of European Working Group on Sarcopenia in Older People (EWGSOP) instead of reference 1. (Sarcopenia: revised European consensus on definition and diagnosis).

Response:

We have shortened the introduction to make it more concise by reorganising sentences and citations. Hence, the paragraph of introduction section is shortened into three paragraph from four. In addition, the number of references reduced to 47 from previously 52 references. We agreed to cite the guideline of The European Working Group on Sarcopenia in Older People (EWGSOP) by Cruz-Jentoft et al. (2010) as it was an original citation. 

The revision highlighted as following:

a. Paragraph one, line 47 (reference 1).

b. Paragraph one, line 55 – 57.

c. Paragraph two, line 60.

d. Paragraph two, line 65 – 67.

e. Paragraph three, line 71 – 76.

3. Material and methods:

a. Add the date and number of ethical approval. 

b. Authors mentioned that Gait speed was determined based on a timed walk on a 2.5-m straight flat surface. Please cite reference for this measurement. Do authors mean 4 or 6 meter walking speed? 

c. What are the inclusion and exclusion criteria of the study?

Response:

a. Datasets that were analysed provided the ethical clearance number obtained from the Institutional Review Board of the Survey METER in Indonesia and Research and Development, Indonesia (RAND) for publication purposes. We presented the protocol approval number in study sample and procedure section, paragraph one, line 95 - 97. 

b. Gait speed was based on the result of a timed walk of 2.5-m straight flat surface. Reference was cited and presented in the sarcopenia section, line 119.

c. Inclusion and exclusion criteria were added in study sample and procedure section, paragraph two, line 98 - 99.

4. Results:

What are the differences between sarcopenic and non-sarcopenic patients according to the PEFR <50%, 50% to 80%, and >80% of the predicted value? You had better write in table1.

Response:

The differences between sarcopenics and non-sarcopenics according to predicted PEFR <50%, 50% to 80%, and PEFR >80% are presented in table 1. 

5. Discussion:

a. You may wish to cite some more recent papers, which support this manuscript’s findings (please see below-references section). 

b. Line 237 and 238 ----- ‘We also demonstrated that handgrip strength was significantly associated with FEV1, maximal voluntary ventilation, and lung function (37, 38).’ Did you demonstrate that in your study?

Response:

a. We appreciate the reviewer’s valuable suggestion to cite recent papers supporting the finding. 

b. We have clarified the statement in previously line 237 and 238 that stated ‘We also demonstrated that handgrip strength was significantly associated with FEV1, maximal voluntary ventilation, and lung function (37, 38).’ Our study did not demonstrate the measurement; it was a citation from the study of Zhu et al. (2019) and Son et al. (2018). The statement was revised to improve the quality of discussion.

6. References:

a. There is no reference from 2020. For an article that will be published this year, the authors may well wish to add some more recent papers (i.e. the very nice article on diaphragmatic muscle thickness and PEFR measurements in sarcopenic patients-Diaphragmatic muscle thickness in older people with and without sarcopenia, PMID: 32406014). 

b. Try to reduce the references. Revise references, many of them are incorrect, including the abbreviations of the journals or non-sense words. See journal information for authors

Response:

a. We have cited a recent paper entitled ‘diaphragmatic muscle thickness in older people with and without sarcopenia, PMID: 32406014’. The paper supported our discussion in paragraph two, line 230 – 231.

b. Some incorrect abbreviations and non-sense words of reference lists were corrected to conform to PLOS ONE reference guideline.

Reviewer #2: 

1. This is an observational study performed in a large sample of community-dwelling Indonesian individuals aged ≥60 years (n = 2422) aimed to analyses prevalence of sarcopenia and its relationship with peak flow rate as a useful biomarker for analysis of sarcopenia. There are some major concerns that need to be addressed in order to support the methodology and thus, the authors' conclusions. Authors need to include and cite all study performed in ASIA about sarcopenia prevalence in older individual only one reference has been cited (page 3 line 61).

Response:

We added a prevalence meta-analysis of sarcopenia in Japanese community-dwelling older population to support previously included evidence, in introduction section, paragraph one, line 55 - 57. 

2. Authors should clarify how sarcopenia criteria were used to diagnose sarcopenia because one individuals fulfilled one or two criteria out 3 what happens?, an algorithm of diagnosis of sarcopenia should be included.

Response:

In our study, sarcopenia was determined based on the diagnosis criteria presented by Asian Working Group for Sarcopenia which categorized sarcopenia into two indicator: sarcopenia and non-sarcopenia. In the method section, sarcopenia part, line 109 - 112 explained the criteria used to diagnose sarcopenia. Algorithm of diagnosis sarcopenia adopted from AWGS was added as Fig. 2. 

3. Sensitivity, specificity and Cronbach alpha data should be stated for all questionnaire used in the study. 

Response:

Cronbach’s alpha values for internal consistency the questionnaire of sleep quality and depressive symptoms were added in the manuscript, in confounders section, paragraph one, line 154 - 157, and line 160 - 162.

4. The criteria used to classify individuals as having "high", "moderate", and "low" physical activity are not explained in details, cut off values (minutes/per week) should be included and referenced properly as it is a crucial variable in this study and in general in the sarcopenia research field.

Response:

The classification of physical activity revised with detail cut of point based on IPAQ guidelines and it was presented in the confounder section, paragraph one, line 149 - 151.

5. Relevant study published on respiratory function and sarcopenia have not been cited, hence the discussion is quite poor: 

a. Landi F, Salini S, Zazzara MB, Martone AM, Fabrizi S, Bianchi M, Tosato M, Picca A, Calvani R, Marzetti E. Relationship between pulmonary function and physical performance among community-living people: results from Look-up 7+ study. J Cachexia Sarcopenia Muscle. 2020 Feb;11(1):38-45. 

b. Martínez-Arnau FM, Buigues C, Fonfría-Vivas R, Cauli O. Respiratory Muscle Strengths and Their Association with Lean Mass and Handgrip Strengths in Older Institutionalized Individuals. J Clin Med. 2020 Aug 24;9(9):2727. 

c. Pacifico J, Geerlings MAJ, Reijnierse EM, Phassouliotis C, Lim WK, Maier AB. Prevalence of sarcopenia as a comorbid disease: A systematic review and meta-analysis. Exp Gerontol. 2020 Mar;131:110801.

Response:

These published articles were important for our study. We appreciate this suggestion. We have cited these articles to improve the quality of our discussion. These articles were cited in the paragraph 2, discussion section, line 226 – 227, line 234– 235, and in paragraph five, line 284 – 286.

---

## [Decision Letter · Decision Letter 1]

15 Jan 2021

Peak expiratory flow rate and sarcopenia risk in older Indonesian people: a nationwide survey

PONE-D-20-24364R1

Dear Dr. Ridwan,

We are very pleased to inform you that your manuscript has now been judged scientifically suitable by our expert Reviewers and it is ready for publication and will be formally accepted for publication once it meets all outstanding technical requirements. I concur with the recommendations of the Reviewers and we will proceed with the publication of your paper.

Best wishes and kind regards,

Stephen E Alway, Ph.D.

Academic Editor

PLOS ONE

Additional Editor Comments (optional):

Reviewers' comments:

Reviewer's Responses to Questions

**Comments to the Author**

1. If the authors have adequately addressed your comments raised in a previous round of review and you feel that this manuscript is now acceptable for publication, you may indicate that here to bypass the “Comments to the Author” section, enter your conflict of interest statement in the “Confidential to Editor” section, and submit your "Accept" recommendation.

Reviewer #1: All comments have been addressed

Reviewer #2: All comments have been addressed

2. Is the manuscript technically sound, and do the data support the conclusions?

Reviewer #1: Yes

Reviewer #2: Yes

3. Has the statistical analysis been performed appropriately and rigorously? 

Reviewer #1: Yes

Reviewer #2: Yes

4. Have the authors made all data underlying the findings in their manuscript fully available?

Reviewer #1: Yes

Reviewer #2: Yes

5. Is the manuscript presented in an intelligible fashion and written in standard English?

Reviewer #1: Yes

Reviewer #2: Yes

6. Review Comments to the Author

Reviewer #1: The authors have addressed all comments I have mentioned. It is a well-design study with data from a nationally representative sample.

Thank you

Kind regards,

Reviewer #2: (No Response)

7. PLOS authors have the option to publish the peer review history of their article (what does this mean?). If published, this will include your full peer review and any attached files.

Reviewer #1: No

Reviewer #2: No

---

## [Editor Report · Acceptance letter]

21 Jan 2021

PONE-D-20-24364R1 

Peak expiratory flow rate and sarcopenia risk in older Indonesian people: a nationwide survey 

Dear Dr. Ridwan:

I'm pleased to inform you that your manuscript has been deemed suitable for publication in PLOS ONE. Congratulations! Your manuscript is now with our production department. 

Kind regards, 

on behalf of

Dr. Stephen E Alway 

Academic Editor

PLOS ONE